# Hypoxia Decreases Diagnostic Biomarkers for Aspergillosis In Vitro

**DOI:** 10.3390/jof5030061

**Published:** 2019-07-11

**Authors:** Elisabeth Maurer, Maria Aigner, Cornelia Lass-Flörl, Ulrike Binder

**Affiliations:** Department of Hygiene, Medical Microbiology and Public Health, Institute of Hygiene and Medical Microbiology, Medical University of Innsbruck, Schöpfstrasse 41, 6020 Innsbruck, Austria

**Keywords:** hypoxia, *Aspergillus* spp., biomarker release, galactomannan, β-(1,3)-glucan

## Abstract

The aim of the study was to evaluate the influence of hypoxia on galactomannan and (1,3)-β-d-glucan release of clinically relevant *Aspergilli* in vitro. Hypoxia decreased biomass and consequently led to lower biomarker release. However, when normalized to biomass, hypoxia led to increased levels of biomarkers at early growth stages (24 h). Antifungals (amphotericin B and voriconazole) decreased the galactomannan amount of *A. fumigatus*, even more prominently in hypoxia.

## 1. Introduction

Besides the patient’s condition, the outcome of invasive aspergillosis depends also on a fast, specific diagnosis and subsequent treatment, both still a challenge in the clinic. One strategy to improve diagnosis is the detection of circulating fungal antigens such as galactomannan (GM) and (1,3)-β-d-glucan (BDG) by commercially available diagnostic kits [1,2,3]. Although these assays have high sensitivity and specificity, variations in diagnostic performance are known [1] but only partially understood. Hypoxia is a microenvironmental stress occurring during pulmonary fungal infections in vivo, and oxygen concentrations at sites of infections can drop as low as 1% [4]. Further, such low oxygen concentrations are known to influence the fungal cell wall composition of *Aspergillus* (*A.*) *fumigatus*, resulting in increased BDG levels in the cell wall [5]. Therefore, hypoxia may contribute to false negative or positive results in GM and BDG diagnostic assays during IPA (invasive pulmonary aspergillosis) [6,7,8,9].

The aim of this study was to compare the GM and BDG release of different *Aspergilli* grown in normoxic or hypoxic conditions. Additionally, we aimed to determine changes in GM release of *A. fumigatus*, the most clinically relevant *Aspergillus* species, in the absence or presence of antifungal agents.

## 2. Materials and Methods

The strain set comprised five clinical isolates of *A. fumigatus*, *A. terreus*, and *A. flavus* each. Hypoxic conditions were set to 1% O_2_, 5% CO_2_, 94% N_2_ (Biospherix C-Chamber & Pro-Ox, Pro-CO2 controller, Parish, NY, USA), and cultures were incubated in normoxia (21% O_2_) in parallel. Fungi were grown for 24, 48, and 72 h in RPMI_1640_ media containing 2% glucose. Mycelia were harvested and lyophilized for dry weight determination. BDG and GM release were determined with a Fungitell^®^ kit (Associates of Cape Cod, Falmouth, MA, USA) and a Platelia™ *Aspergillus* AG kit (Biorad, Marnes la Coquette, France). Supernatants were diluted 1:10,000 (GM) or 1:100 (BDG), respectively, and assays were performed according to the manufacturer’s instructions. Medium in equal dilutions as the samples was included as a negative control and did not result in positive GM or BDG values at the dilutions tested. Where applicable, values were normalized to biomass (dry weight). For the kinetic analysis of GM release, strains were pre-grown in normoxia for 16 h, and equal amounts of biomass (wet weight) were shifted to hypoxia, or kept at normoxia, respectively. Amphotericin B (0.5 µg/mL; Bristol Meyer Squibb, Austria) or voriconazole (0.125 µg/mL; Pfitzer, Ltd., Sandwich, UK), at concentrations based on the MIC (minimal inhibitory concentration) of the test strains, were added to the cultures simultaneously. Aliquots of culture supernatants were sampled at 4, 8, 12, 24, and 48 h. All experiments were performed in biological triplicates. Statistical analysis was done using GraphPad Prism 6 software (San Diego, CA, USA). Biomarker levels of samples grown in hypoxia were compared to normoxia using multiple *t*-test (Mann–Whitney). For kinetic time course measurements, significance was calculated using a two-way ANOVA. *p* ≤ 0.05 were regarded as statistically significant.

## 3. Results and Discussion

As hypoxia led to impaired growth of all tested *Aspergilli* (Appendix A), GM and BDG amounts in supernatants shown in Figure 1 were normalized to biomass. Hence, significantly (*p* ≤ 0.05) increased GM and BDG release was measured at early growth stages (24 h) in hypoxic cultures of *A. fumigatus* and *A. terreus*, whereas no significant differences were determined at later time points (Figure 1, Appendix A). Although the increase in both biomarkers was strong in the *A. flavus* cultures grown under hypoxic conditions (1.8-fold for GM and 3.6-fold for BDG), the values did not reach statistical significance according to our definition (GM: *p* = 0.23, BDG: *p* = 0.13), a fact mainly due to the high standard deviation in the hypoxic cultures. The amount of released biomarker differed between the three tested *Aspergillus* species, with *A. terreus* showing the highest average values of GM and BDG per mg dry weight in both conditions and at all time points except for *A. fumigatus* at 24 h. The presence of antifungals (amphotericin B (AMB) and voriconazole (VOR)) led to decreased GM levels in both oxygen conditions. Differences were observed from 12 h post-treatment onwards but reached statistically significant differences only at 24 and 48 h. The combination of hypoxia and antifungal treatment led to even more pronounced reduction in GM release compared to cultures grown in normoxia (Figure 2). At 24 h, the GM index of untreated normoxic cultures (=100%, 5525 ± 226) dropped to 58% in AMB-treated cultures (3228 ± 496, *p* = 0.04) and 77% in VOR treated samples (4272 ± 888, *p* = 0.5). Incubation in hypoxic conditions aggravated this reduction, leading to a reduced GM index of 41% in AMB cultures (2439 ± 738; *p* = 0.002) and 44% in VOR cultures (2280 ± 770, *p* = 0.001).

The reliability of commercial kits detecting fungal antigens in patient samples was shown to be influenced by many biological and epidemiological factors [10]. Hypoxic microenvironmental conditions during IPA are attributed to tissue damage through hyphal invasion accompanied by the resulting inflammatory response and inhibition of angiogenesis by the fungus [4,11,12]. Our results are in agreement with the findings of Brock et al. with the aid of bioluminescence [13]. Using an oxygen-dependent luciferase-producing *A. fumigatus* strain, a maximum of luminescence was reached after 24 h indicating swelling and germination of conidia accompanied by the higher amount of biomarkers detected at 24 h in our study. Subsequently, because of tissue damage and reduced oxygen availability, bioluminescence decreased. The following growth stagnation might explain reduced biomarker detection at later time points, as GM is released into the surrounding media mainly during active growth and hyphal extension [14]. Hypoxia was shown to cause modifications in cell wall components of *A. fumigatus* [5]. The observed increase of glucan in the cell wall in answer to hypoxia, is reflected by our data indicating increased release of BDG at early time points of in vitro cultures. Further, our data indicate that these changes were not significant at later time points and might therefore not contribute to false positive results during IPA.

In our study, the combination of hypoxic environment and antifungal treatment led to even more pronounced reduction of GM release than antifungal drugs alone. Antifungal treatment was shown to influence biomarker release and detection in vivo [15,16,17] and in vitro, where decreased GM values were attributed to a reduction in mycelia mass [1]. GM values were found to be almost zero following treatment with amphotericin B, correlating with less or no growth, while itraconazole had no influence on growth ability and GM release [18]. The pronounced decrease in GM release following antifungal treatment in combination with impaired growth during hypoxia might possibly explain false-negative results obtained with the GM assay. Our results might be considered especially for patients receiving anti-mold prophylaxis known to exhibit a multitude of negative results and if the detection of biomarkers is a suitable tool to assess treatment response [17,19].

## Figures and Tables

**Figure 1 jof-05-00061-f001:**
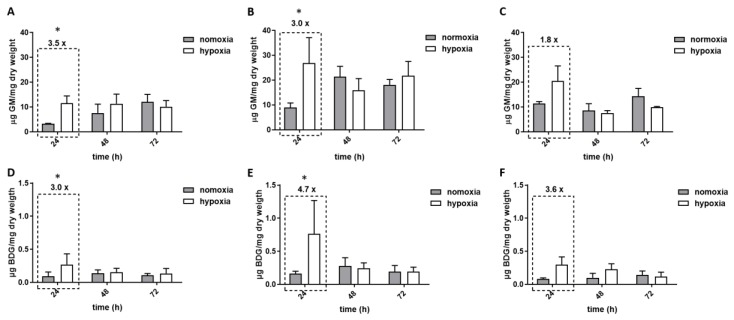
Galactomannan (GM) and beta-glucan (BDG) content in supernatant of clinically relevant aspergilli. *A. fumigatus* (**A**,**D**), *A terreus* (**B**,**E**), and *A. flavus* (**C**,**F**) cultures were grown in hypoxic conditions (white bars) and normoxic conditions (grey bars). The amount of GM and BDG was normalized to biomass (dry weight). Numbers above boxes indicate fold change of biomarkers released in hypoxia in comparison to normoxia, error bars indicate standard deviation, asterisks (*) indicate statistical differences (multiple *t*-test) between normoxic and hypoxic values.

**Figure 2 jof-05-00061-f002:**
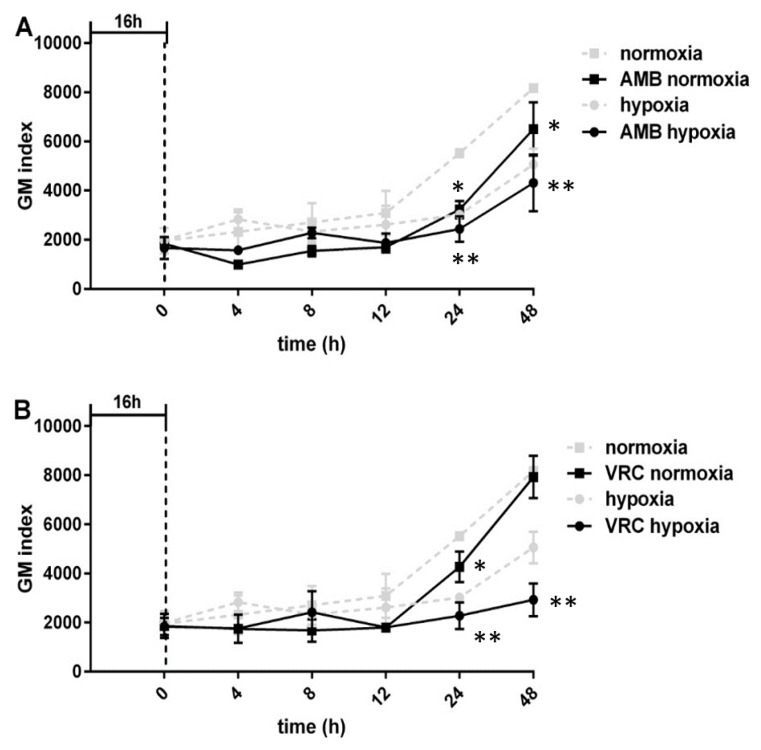
Kinetics of GM release by *A. fumigatus* indicated as an index in the absence (broken grey lines) and presence (unbroken black lines) of (**A**) AMB and (**B**) VRC under normoxic (■) and hypoxic conditions (●). Vertical lines define the shift to either normoxia or hypoxia, and subsequent addition of antifungals (16 h; nomoxic conditions). Stars (*) indicate statistically significant differences of antifungal-treated samples compared to the normoxic control: * indicates *p* ≤ 0.05, ** indicates *p* ≤ 0.005.

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
