# Peer review of "Hypoxia Decreases Diagnostic Biomarkers for Aspergillosis In Vitro"

_jof, 2019, doi:10.3390/jof5030061_

Round 1

Reviewer 1 Report

The authors described the effect of hypoxia and antifungal exposure on the growth of Aspergillus species and correlate this with the expression of galactomannan (GM) and b-D-glucan (BDG).

1. Why was 1% O2 chosen for the hypoxia condition instead of a level more typical for hypoxia in humans? This seems artificially low and it is not clear how results obtained at this level will correlate with a more physiologic level.

2. Figure 1 and Supplemental Figure S1 appear to be the same? On line 49, figure 1 is reported to show GM and BDG amounts under normoxia and hypoxia conditions but that is not shown in figure 1. Figure 1 shows how extreme hypoxia influences the growth rate of Aspergillus species. Please confirm that the figures present the correct data.

3. Line 52 indicates that A. terreus had the highest levels of GM and BDG of the Aspergillus species tested but Table S1 indicates that A. terreus and A. flavus levels were overlapping in many instances if the standard deviation of the measurements was taken into consideration. Please revise line 52.

4. Line 54 indicates that antifungals (amphotericin and voriconazole) led to decrease GM levels but Figure 2 indicates that the GM levels overlapped in the presence and absence of antifungals for the early timepoints (0-12 hrs) and had some overlap (for hypoxia +/- amphotericin and for normoxia +/- voriconazole) out to 48 hours. The GM index increased after 24-48 hrs but the authors do not have a control without antifungal and they do not address the potential for antifungal degradation over time in the test medium so it is difficult to draw a conclusion about the significance of the measured GM levels over time. Please revise line 54.

5. Line 69 states that the data presented indicates that BDG was not liberated during IPA but it is not clear what data is presented to support this conclusion. Table S1 seems to indicate that BDG amounts increased under hypoxic conditions. Please clarify which figure or table describes the lack of liberation of BDG.

Author Response

The authors described the effect of hypoxia and antifungal exposure on the growth of Aspergillusspecies and correlate this with the expression of galactomannan (GM) and b-D-glucan (BDG).

1. Why was 1% O2 chosen for the hypoxia condition instead of a level more typical for hypoxia in humans? This seems artificially low and it is not clear how results obtained at this level will correlate with a more physiologic level.

è This concentration of oxygen was chosen because it has been stated by others (see references 4, 5) that in sites of Aspergillus-infections, such as the lung, oxygen can drop to levels of 1% or even lower. We have included this information in the revised version. Also, in previous studies carried out in our laborator we used the same concentration of oxygen fort he same reasons (Effect of reduced oxygen on the antifungal susceptibility of clinically relevant aspergilli. Binder U, Maurer E, Lackner M, Lass-Flörl C. Antimicrob Agents Chemother. 2015 Mar;59(3):1806-10;

Susceptibility profiles of amphotericin B and posaconazole against clinically relevant mucorales species under hypoxic conditions. Maurer E, Binder U, Sparber M, Lackner M, Caramalho R, Lass-Flörl C. Antimicrob Agents Chemother. 2015 Feb;59(2):1344-6.

2. Figure 1 and Supplemental Figure S1 appear to be the same? On line 49, figure 1 is reported to show GM and BDG amounts under normoxia and hypoxia conditions but that is not shown in figure 1. Figure 1 shows how extreme hypoxia influences the growth rate of Aspergillus species. Please confirm that the figures present the correct data.

è We are thankful for this comment, there has been an error made when introducing the figures tot he text. Figure 1 and S1 should not be the same – we ade changes accordingly. Now Figure 1 presents biomarker levels measured in the supernatant, and Figure S1 presents the effect of hypoxia on the growth/biomass of the aspergilli.

3. Line 52 indicates that A. terreus had the highest levels of GM and BDG of the Aspergillusspecies tested but Table S1 indicates that A. terreus and A. flavus levels were overlapping in many instances if the standard deviation of the measurements was taken into consideration. Please revise line 52.

è We analysed the differences oft he average amounts oft he biomarkers and have made changes to the text accordingly – please refer to line 60, where now we state „Amount of released biomarker differed among the three tested Aspergillus species, with A. terreus showing the highest average values of GM and BDG per mg dry weight at both conditions and at all time points except for A. fumigatus at 24 h.“  

4. Line 54 indicates that antifungals (amphotericin and voriconazole) led to decrease GM levels but Figure 2 indicates that the GM levels overlapped in the presence and absence of antifungals for the early timepoints (0-12 hrs) and had some overlap (for hypoxia +/- amphotericin and for normoxia +/- voriconazole) out to 48 hours. The GM index increased after 24-48 hrs but the authors do not have a control without antifungal and they do not address the potential for antifungal degradation over time in the test medium so it is difficult to draw a conclusion about the significance of the measured GM levels over time. Please revise line 54.

è The control without antifungals are shown in Figure 2 in grey, dotted lines, as indicated in the figure legend. We have revised the text concerning antifungal degradation and included values of GM index plus statistics.

5. Line 69 states that the data presented indicates that BDG was not liberated during IPA but it is not clear what data is presented to support this conclusion. Table S1 seems to indicate that BDG amounts increased under hypoxic conditions. Please clarify which figure or table describes the lack of liberation of BDG.

è We agree with this comment and have rephrased the discussion part from line 81 on. We agree with the reviwer that BDG increased in the early time points, which correlates to the data shown in the cited study but has no effect on later time points.

Reviewer 2 Report

1.       For the clinical isolates, ethical approval is missing

2.       Line 61, check the typo-error after the reference (13)

3.       Homogenize, at places for hours it is written ‘h’ or ‘hours’

4.       Lines 50-51, the authors state there was no significant differences in the GM/BG levels in the later stages of growth (Table S1), but the GM values are not convincing; could the authors provide exact statistical significance values for these?

Author Response

 For the clinical isolates, ethical approval is missing

è Sampling of clinical isolates was done according to the ethical agreements of the MUI and Austria. In Austria, there is no need of ethical approval for in vitro fungal culture experiments.

2.       Line 61, check the typo-error after the reference (13)

è We have made changes accordingly.

3.       Homogenize, at places for hours it is written ‘h’ or ‘hours’

è We have made changes accordingly.

4.       Lines 50-51, the authors state there was no significant differences in the GM/BG levels in the later stages of growth (Table S1), but the GM values are not convincing; could the authors provide exact statistical significance values for these?

è We have included the statistical analysis (Multiple t test) and the p values in table 1 and ist legend in the revised version of the manuscript.

Reviewer 3 Report

This is a short communication and therefore, the sections are condensed with no subheadings. The introductory paragraph is concise, explaining the clinical problem and aims are set out. Methods appear sound. There are a few things to highlight:

L36 respectively is not needed

L37 - 38 Explanation of the medium, is this the negative control? If so, say so. 

Were the antifungal concentrations administered taken from a reference? 

L48 Figure S1 and Fig 1 appear to be the same.  and neither show impaired growth as is described in the text.

L55 - 56 this line states that GM levels were decreased even more so in hypoxic conditions than normoxic conditions in the presence of antifungals. I do not see this on the graphs described except in one case with VRC at 48h. Please explain. 

L62 there is a large gap between the words strain and a maximum

The results in Fig 2 only show GM index and there are no beta-glucan results. The abstract and aims included beta-glucan. Please explain. 

Overall conclusions good, with emphasis on clinical practice and consequences for patients. 

Author Response

This is a short communication and therefore, the sections are condensed with no subheadings. The introductory paragraph is concise, explaining the clinical problem and aims are set out. Methods appear sound. There are a few things to highlight:

L36 respectively is not needed

è We have made changes accordingly.

L37 - 38 Explanation of the medium, is this the negative control? If so, say so. 

è Yes, we have included this information in the text. Line 38 and 39

Were the antifungal concentrations administered taken from a reference? 

ð  They were chosen according tot he MIC of the test strains. We included this information in the text. Line 43

L48 Figure S1 and Fig 1 appear to be the same.  and neither show impaired growth as is described in the text.

ð  We have changed the figures accordingly in the revised version. Figure 1 now shows the amount of biomarkers released normalized to biomass, while Figure S1 shows the reduced growth of the respective species under hypoxic conditions. We do not understand, why the reviewer stated that neither oft he figures shows impaired growth – in figure S1 the percentage of growth normalized to normoxic conditions in shown – e.g. cultures of A. fumigatus grown in hypoxia only reached little more than 20% of the biomass of cultures grown in normoxic conditions.

L55 - 56 this line states that GM levels were decreased even more so in hypoxic conditions than normoxic conditions in the presence of antifungals. I do not see this on the graphs described except in one case with VRC at 48h. Please explain. 

ð  We have adapted the text oft he revised version of the manuscript accordingly, to make this difference more visible, including values oft he GM index, standard deviation and the p- value of 2 way anova analysis. Please refer to line 61 ff.

L62 there is a large gap between the words strain and a maximum

è We have made changes accordingly.

The results in Fig 2 only show GM index and there are no beta-glucan results. The abstract and aims included beta-glucan. Please explain. 

ð  We changed figure 1 to the original figure we wanted to shown – clearly showing both GM and BDG results. This misunderstanding is based on our error in the first version, where figure 1 and S1 was mixed up. Also table S1 shows both biomarkers.

ð  We have made changes to the text explaining the aims of our study in more detail, please see line 30, 31.

Overall conclusions good, with emphasis on clinical practice and consequences for patients

Reviewer 4 Report

The authors have written an article on the release of GM and BG of 3 different Aspergillus species (A. fumigatus, A. terreus, and A. flavus) during hypoxic and antifungal treatment stress in vitro.

Although the topic of this work is interesting, my main concern in publishing this work is, whether the conclusions the authors extract can really be extracted from the results. I would recommend the changes listed below.

In the title the authors write that hypoxia decreases diagnostic biomarkers for aspergillosis. In fact, in the paper only results for galactomannan are presented. Additionally, the authors should indicate in the title that in vitro studies were used to draw their conclusions. The title has to be changed.

In the introduction I do not really understand how reference 6 fits into lines 26-27. In line 28-29 the aim of the study is presented. The authors want to compare the GM and BDG release of different Aspergilli grown in normoxic or hypoxic conditions in the absence or presence of antifungal agents. In fact, kinetics in normoxia or hypoxia, with or without antifungals are only presented for GM and additionally only for A. fumigatus. So, the paper does not handle the main questions of the study aim.

In line 49 is a major problem in the results, as the text does not fit to the figure. In my version of the peer review article Figure 1 and Figure S1 are the same. This has to be changed. I would delete the sentence about intra-species variation in line 53. I do not think that 5 strains are sufficient to draw conclusions about intra-species variations, especially with regard to the relatively high standard derivation in some instances.

For me the global presentation of the results is not very clear. In the materials and methods section the use of statistical analysis is noted, but no statistics are presented in the results section. It would recommend to include p values in Figure 2 and Table S1. This would improve the drawing of a conclusion of the results significantly.

My major concern on this study is the drawing of the conclusion in line 76-79. It is clear that there is a difference between GM release in normoxic and hypoxic conditions, but for example in normoxic conditions there is no difference between normoxia alone and normoxia + voriconazole. Additionally, to my opinion (as we do not have statistics) there is no difference between hypoxia and hypoxia + AMB. Therefore, the conclusion has to be changed or results have to be presented which prove the conclusion.

For beta 1,3 glucan not always the same abbreviation is used throughout the paper (text vs. Table S1), please change this.

Author Response

The authors have written an article on the release of GM and BG of 3 different Aspergillus species (A. fumigatusA. terreus, and A. flavus) during hypoxic and antifungal treatment stress in vitro.

Although the topic of this work is interesting, my main concern in publishing this work is, whether the conclusions the authors extract can really be extracted from the results. I would recommend the changes listed below.

In the title the authors write that hypoxia decreases diagnostic biomarkers for aspergillosis. In fact, in the paper only results for galactomannan are presented. Additionally, the authors should indicate in the title that in vitro studies were used to draw their conclusions. The title has to be changed.

ð  We changed the title accordingly to „Hypoxia decreases diagnostic biomarkers for aspergillosis in vitro.

ð  We habe shown both, GM and BDG amounts in supernatants – new figure1 and also in table S1, therefore, we did not change „biomarker“ to only GM

In the introduction I do not really understand how reference 6 fits into lines 26-27. In line 28-29 the aim of the study is presented. The authors want to compare the GM and BDG release of different Aspergilli grown in normoxic or hypoxic conditions in the absence or presence of antifungal agents. In fact, kinetics in normoxia or hypoxia, with or without antifungals are only presented for GM and additionally only for A. fumigatus. So, the paper does not handle the main questions of the study aim.

ð  We are thankful for the comment on Reference 6, as this reference was placed there by misktake. We have changed Reference 6 by the appropriate one, Park et. al, highlighted in the reference list.

ð  We have changed the aim of the study accordinlgy to make it more clear to the readers what we actually achieved. Please see line 30,31.

In line 49 is a major problem in the results, as the text does not fit to the figure. In my version of the peer review article Figure 1 and Figure S1 are the same. This has to be changed. I would delete the sentence about intra-species variation in line 53. I do not think that 5 strains are sufficient to draw conclusions about intra-species variations, especially with regard to the relatively high standard derivation in some instances.

ð  We agree with this comment – the figures have been changed, as the sam efigure was submitted in the first version oft he manuscript, this is now Figure S1, and Figure 1 represents amount of both GM and BDG in the supernatant of all 3 species, normalized to biomass.

ð  We deleted the sentence about intra-species variations.

For me the global presentation of the results is not very clear. In the materials and methods section the use of statistical analysis is noted, but no statistics are presented in the results section. It would recommend to include p values in Figure 2 and Table S1. This would improve the drawing of a conclusion of the results significantly.

My major concern on this study is the drawing of the conclusion in line 76-79. It is clear that there is a difference between GM release in normoxic and hypoxic conditions, but for example in normoxic conditions there is no difference between normoxia alone and normoxia + voriconazole. Additionally, to my opinion (as we do not have statistics) there is no difference between hypoxia and hypoxia + AMB. Therefore, the conclusion has to be changed or results have to be presented which prove the conclusion.

For beta 1,3 glucan not always the same abbreviation is used throughout the paper (text vs. Table S1), please change this.

ð  We have done so accordingly and used BDG as appreviation in all cases.

Round 2

Reviewer 1 Report

All of my concerns have been satisfactorily addressed.

Author Response

thank you

Reviewer 4 Report

Dear authors,

thank you very much for your changes on the manuscript. The manuscript is much better now. I have some minor remarks:

Line 128

Please change BG to BDG in table S1

Figure 2

Please indicate statistical differences in the figure and in the figure legend by, for example, stars, squares, ...

Author Response

We have changes BD to BDG in line 128 (below the figure)

and we have added " * " as a symbol for statistical difference to Figure 2 and explained accordingly in the figure legend for what " * " or "**" stands ( * = p values 0.05 or smaller, ** is p values 0.005 or smaller)

changes are highlighted in yellow